# *MAOA uVNTR* Genetic Variant and Major Depressive Disorder: A Systematic Review

**DOI:** 10.3390/cells11203267

**Published:** 2022-10-17

**Authors:** Ana Beatriz Castro Gonçalves, Caroline Ferreira Fratelli, Jhon Willatan Saraiva Siqueira, Ligia Canongia de Abreu Cardoso Duarte, Aline Ribeiro Barros, Isabella Possatti, Maurício Lima dos Santos, Calliandra Maria de Souza Silva, Izabel Cristina Rodrigues da Silva

**Affiliations:** 1Pharmacy Course, Faculty of Ceilândia, University of Brasília (UnB), Brasília—Federal District (DF), Brasília 72220-900, Brazil; 2Postgraduate Program in Health Sciences and Technologies, Faculty of Ceilândia, University of Brasília (UnB), Brasília—Federal District (DF), Brasília 72220-900, Brazil; 3Clinical Analysis Laboratory, Molecular Pathology Sector, Pharmacy Department, Faculty of Ceilândia, University of Brasília (UnB), Brasília—Federal District (DF), Brasília 72220-900, Brazil

**Keywords:** Major Depressive Disorder, Monoamine Oxidase A, *MAOA uVNTR*, genetic polymorphism, pharmacogenetics, risk-factors

## Abstract

Major Depressive Disorder (MDD) is a highly prevalent multifactorial psychopathology affected by neurotransmitter levels. Monoamine Oxidase A (MAOA) influences several neural pathways by modulating these levels. This systematic review (per PRISMA protocol and PECOS strategy) endeavors to understand the *MAOA uVNTR* polymorphism influence on MDD and evaluate its 3R/3R and 3R* genotypic frequencies fluctuation in MDD patients from different populations. We searched the Web of Science, PubMed, Virtual Health Library, and EMBASE databases for eligible original articles that brought data on genotypic frequencies related to the *MAOA uVNTR* variant in patients with MDD. We excluded studies with incomplete data (including statistical data), reviews, meta-analyses, and abstracts. Initially, we found 43 articles. After removing duplicates and applying the inclusion/exclusion criteria, seven articles remained. The population samples studied were predominantly Asians, with high 3R and 4R allele frequencies. Notably, we observed higher 3R/3R (female) and 3R* (male) genotype frequencies in the healthy control groups and higher 4R/4R (female) and 4R* (male) genotype frequencies in the MDD groups in the majority of different populations. Despite some similarities in the articles analyzed, there is still no consensus on the *MAOA uVNTR* variant’s role in MDD pathogenesis.

## 1. Introduction

Major Depressive Disorder (MDD) is a psychiatric condition marked by various mood changes (e.g., sad, empty, or irritable) and reduced interest and pleasure for at least two weeks, among other varying symptoms. Considered one of the most prevalent mood disorders [1,2], the intensity of symptoms can be disabling with clinically significant suffering [3]. The high rates of first-line treatment remission (30–40%) have patients often undergoing multiple subsequent courses of antidepressants or augmentation strategies, making MDD a formidable public health challenge for front-line clinicians and researchers [4].

Worldwide more than 350 million people were estimated to suffer from MDD in 2017 [3,5], and by the year 2030, MDD should become the highest global burden of disease, according to the World Health Organization (WHO) [1,6,7]. This burden would represent not only a critical compromise of the affected population’s quality of life but also a significant economic impact—even without not considering the impact of the COVID-19 pandemic on the disorder. The COVID-19 pandemic triggered a 25% increase in depression (*p* = 0.029) and anxiety (*p* = 0.0001) prevalence worldwide in just its first year [8]. Its actual impact on the world’s mental health is still to be determined [9,10].

The mechanisms for developing MDD are not well understood as the cause seems to involve a complex interaction of several factors, including biological (genetic/physiological) and psychosocial (environmental) [7]. Among the various theories to explain this disease onset, the most cited is the monoaminergic hypothesis [11], in which the scarcity of neurotransmitters, such as serotonin and norepinephrine in the synaptic cleft contributes significantly to symptomatological manifestations. However, this theory does not explain all aspects of the disease, reinforcing the multifactorial aspect of its etiology [12]. Several genetic and environmental models have been proposed to relate these diverse aspects to MDD etiology.

The Monoamine Oxidase A (MAOA) gene, located in the p11.23 region of the X chromosome, is an attractive candidate for genetic studies due to its influence on several neural pathways [13]. As *MAOA* is an X-linked gene, males are considered hemizygotes (have only one allele, e.g., A* or B*, with * indicating the missing chromosome) for the gene, given that males only have one X chromosome, whereas females can be homozygotes (have two of the same allele, e.g., A/A) or heterozygotes (have two different alleles, e.g., A/B) for the gene. This gene is responsible for producing the MAOA mitochondrial outer membrane enzyme isotype that acts in the oxidative catalysis of monoamines in the synaptic cleft, regulating the brain’s monoamine levels [13]. The MAOA enzyme has a higher affinity for hydroxylated amines, such as serotonin, dopamine, and noradrenaline [6], and is mainly expressed in catecholaminergic presynaptic neurons [14]. Dysfunctions in this enzyme function or levels may lead to neuropsychiatric disorder pathogenesis and clinical manifestations, including depression [15]. Consequently, the *MAOA* genetic variants (polymorphisms) that alter its gene transcriptional activity or enzyme function/activity may contribute to an imbalance in the monoaminergic neurotransmission system [16].

Sabol et al. [17] identified the *MAOA* upstream variable number of tandem repeats (*uVNTR*) functional polymorphism in the MAOA gene’s promoter region that seems to affect its transcriptional [17,18] and enzyme activity [18], as measured in blood or brain [18], but not its mRNA abundance [18]. *MAOA uVNTR* polymorphism also does not seem to affect its gene’s methylation [18]. This polymorphism consists of repeating sequences of 30 base pairs—2, 3, 3.5, 4, 5, or 6 repeats (R). The 3R (also known as 1) has low transcriptional efficiency and is classified as a low-activity allele (sometimes also called short-form alleles), whereas the presence of the 3.5R (also known as 2) and 4R (also known as 3) alleles leads to a more efficient transcriptional activity and, hence, they are classified as high-activity alleles (sometimes also called long-form alleles) [17]. There is still no consensus regarding the other alleles’ transcriptional influence; although some consider 2R and 5R (also known as 4) low-activity alleles [16,17,19,20], others consider them high [21,22,23,24,25,26].

Considering the *MAOA uVNTR* variant’s influence on the MAOA enzyme’s transcriptional and enzymatic activity and the role of monoamine deficiency in MDD pathogenesis, it is theorized that the 3R low-activity allele may act as a protective factor against MDD [27]. Nevertheless, the evidence on the *MAOA uVNTR* variant’s influence on MDD development is limited and divergent. The present systemic review evaluated the *MAOA uVNTR*’s 3R/3R (female homozygote) and 3R* (male hemizygote) genotypic frequency fluctuations in MDD studies conducted in different populations.

## 2. Materials and Methods

### 2.1. Search Strategy and Selection Criteria

The present systematic review was developed following the Preferred Reporting Items for Systematic Reviews and Meta-Analyses (Prisma) guidelines, pending PROSPERO number due to COVID-19 reviews prioritization (registered on the 10th of April 2022, CRD42022324909). Our inclusion criteria were based on Population, Exposure, Comparison, Outcome, and Study type (PECOS) strategy, considering: (1) population: human research participants with Major Depressive Disorder (MDD); (2) exposure: *MAOA uVNTR* genetic variant; (3) comparison: the *MAOA uVNTR* variant’s 3R/3R (female homozygote) and/or 3R* (male hemizygote) genotypic frequencies; (4) outcome: 3R/3R and/or 3R* genotypic frequencies fluctuation in different populations; (5) study type: observational and interventional.

For this, we included observational or interventional studies that presented data on the *MAOA uVNTR*’s variant genotypic and allelic frequencies in human research participants with MDD and described their laboratory methods according to our eligibility criteria. However, studies with incomplete data (including statistical data), reviews, meta-analyses, abstracts, and studies not in English, Spanish, or Portuguese were excluded.

We searched, on the 11th of May 2022, employing the databases Web of Science, PubMed, Virtual Health Library (BVS), and EMBASE with no adopted filters, including the year of publication of the articles. The indexed terms (descriptors) researched were “MAOA uVNTR,” “Depressive Disorder, Major,” and “Polymorphism, Genetic,” the last two as described by the Medical Subject Headings (MeSH) vocabulary thesaurus, combined by the boolean operator “AND.”

### 2.2. Study Selection and Data Extraction

Two reviewers (CF and AG) collaborated on the article selection in two phases. Each reviewer in the first phase independently analyzed each article’s title and abstract, verifying their eligibility according to the PECOS strategy. The Rayyan tool, developed by the Qatar Computing Research Institute (QCRI), was used to assist this initial analysis, while Mendeley Desktop version 1.19.4 software helped remove duplicates. In the second phase, the same two reviewers (CF and AG) independently performed the full-text analysis of the pre-selected articles, always observing the pre-established eligibility criteria. For this, Mendeley Desktop software version 1.19.4 was also used.

In both phases, disagreements or doubts were discussed between the two reviewers, and a third reviewer (CS) was consulted if a disagreement was unresolved. The two reviewers (CF and AG) independently extracted pre-defined data into an electronic spreadsheet in Microsoft Office Excel; these were: author, study title, objective, year of publication, the country in which the study was carried out, variants studied, 3R/3R and 3R* genotypic frequencies, sample size, laboratory methodology, main result, and *p*-value. Relevant or other up-to-date original publications in the field have also been included in the introduction and discussion section.

### 2.3. Bias Risk in Each Study

Risk models are usually based on either examining genetic variants or analyzing genetic and environmental risk factors. We examined the selected articles’ bias risk using the Genetic Risk Prediction Studies (GRIPS) Guideline [28,29], comprised of 25 items, to rate the articles’ quality. This review only considered 20 items (verifying the item’s presence or absence) when evaluating the studies’ quality. We considered that an article was of good quality if it presented at least 75% of the items.

This step was performed independently by two reviewers (CF and AG). Any disagreements were resolved after discussing with the third reviewer (IS).

## 3. Results

### 3.1. Articles’ Search, Selection, and Quality Assessment

Initially, we identified 43 articles by searching four databases. After removing duplicates, 20 studies were selected for the title and abstract analysis, observing the aspects delimited by PECOS. Among these, 11 were selected for full-text analysis. After considering previously established inclusion and exclusion criteria, seven articles were eligible for inclusion in this systematic review (Figure 1, Table 1).

Articles that did not meet the eligibility criteria (the PECOS strategy) were excluded, and their reason for exclusion is described in Appendix A.

Appendix A presents the results of the selected articles’ bias risk analysis and quality determination using the Genetic Risk Prediction Studies (GRIPS) Guideline (20 items and two sub-items of 25 items). Of articles analyzed in this study, 57.1% (4) had at least 75% (16.5) or more items and were considered of good quality [26,27,30,33]. The lowest score was 68.2% (15 items) [36] Appendix A summarizes the selected articles’ selection criteria for MDD and control groups, the participants’ ethnicity/race, and the articles’ employed statistical analysis.

### 3.2. Selected Studies’ General Characteristics

As shown in Table 2, most studies were conducted on the Asian continent, mainly in Taiwan [27,30], followed by the American and European continents. Regarding demographic data, most studies were conducted with adults over 18 years old, and females were more frequent within the MDD patient groups.

### 3.3. MAOA uVNTR Genotypic Frequency

In general, the *MAOA uVNTR* variant’s high-activity 4R allele had the highest frequency in the populations studied, followed by the low-activity 3R allele. While the 2R, 3.5R, 5R, and 6R alleles, in most studies, were considered rare, being found in few patients or not found at all [23,26,27,30,33,34]. This finding confirms Sabol et al.’s [17] report that the 3R and 4R alleles are the most common among distinct ethnic populations. Taking into account the 3R/3R (female homozygote) and 3R* (male hemizygote) genotype frequencies reported by the articles included in this systematic review, we noted that in all studies, their individual and combined frequencies were lower than 60% in populations diagnosed with MDD [23,26,27,30,33,34] (Table 1 and Table 2).

Du et al. [23] found a combined 3R/3R and 3R* genotypic frequency of 37.2%, similar to Yu et al. [33] who estimated 35.1%, while Lung et al. [27], in turn, found a 41.2% frequency and Sanabrais-Jiménez et al. [36] a 26.6% frequency. The highest frequency estimated was 47.3% by Huang et al. [30], in contrast, the lowest frequency was 13.1%, estimated by Rivera et al. [26]. Won et al. [34] found a 41.93% 3R/3R frequency in the female MDD patients they analyzed (Table 1 and Table 2). The lower percentage of the 3R/3R and 3R* genotypes in the majority of MDD groups might indirectly indicate a protective role against MDD, depending on the population (Table 1 and Table 2).

## 4. Discussion

### 4.1. MAOA uVNTR and Its Genotypic Frequency in Major Depressive Disorder (MDD)

Different factors can influence genotypic frequency in different populations and study groups. Among them, we highlight population stratification and ethnicity, sample size, different symptom detection and diagnosis methodologies applied to case and control groups and also between different studies, phenotype definition (case group), unscreened control groups, multiallelic gene organization, and variations related to sex, e.g., the *MAOA* gene is located on the X chromosome (X-inactivation process also called lyonization and males only have one allele), or even hormonal effects [26,30,33,34]. Some might explain the significant variation in *MAOA uVNTR* polymorphism genotype frequencies between the analyzed populations. In the studies conducted in an Asian population, female patients diagnosed with Major Depressive Disorder (MDD) and a 3R/3R (female homozygote) genotype had frequencies that varied between 25 and 42%, while in male MDD patients with a 3R* (male hemizygote) genotype, the frequency varied from 48 to 58% (Table 1 and Table 2). In contrast, in Caucasians/Hispanics/Latino, these frequency values ranged between 11 and 35% and from 26 to 44%, respectively (Table 1 and Table 2).

Among the studies conducted with Asian populations, Yu et al. [33] found the lowest 3R/3R genotype (25.6%) frequency distribution of the MDD groups. This lower frequency could be due to their study’s small female MDD sample size (*n* = 34), although Won et al. [34] found a 41.9% 3R/3R frequency in the 13 female MDD patients they analyzed. Interestingly, Yu et al.’s [33] control group also had the highest 3R/3R genotype frequency (44.5%), and among MDD patients, the highest 3R/4R genotype frequency (57.1%) within the Asian studies [19]. Similar to Guitiérrez et al. [25] in Spain and Kersting et al. [38] in Germany, female 4R high-activity allele carriers were common in the MDD group (Chi-square Test, X^2^ = 9.65, df = 2, *p* = 0.008) [19]. Same with male 4R* carriers (X^2^ = 6.27, df = 1, *p* = 0.015) [19]. In turn, Lung et al. [27] and Huang et al. [30] had higher frequencies for the combined 3R/3R and 3R* genotypes (41.2% and 47.3%, respectively). Likewise, in these studies, the 3R/3R and 3R* genotype frequency was also usually higher in the control group, except for Huang et al.’s [30] 3R/3R genotype, compared to the MDD patients’ groups (Table 1 and Table 2). The 3R allelic frequencies for the control groups were comparable between studies, ranging from 52% to 67.4% [27,30,33,34]. These frequencies were consistent with those estimated by Kunugi et al. [39] in a Japanese population with unipolar depression (60%).

In Rivera et al.’s [26] study, the combined 3R/3R and 3R* genotype frequency in a Spanish population with MDD was estimated at 13.1%, similar to another Spanish study with MDD patients (16.8%) [25]. Notably, the Rivera et al. [26] study used two instruments for diagnosis, the Diagnostic and Statistical Manual of Mental Disorders—Fourth Edition (DSM-IV) and the Composite International Diagnostic Interview (CIDI) per the International Classification of Diseases—10^th^ edition (ICD-10), with separate analyses for the different phenotypes (ICD-10 depressive episode; ICD-10 severe depressive episode and DSM-IV major depression, MDD). Furthermore, the authors grouped the *MAOA uVNTR* variants’ high-activity 3.5R and 4R alleles, the 5R allele, and their combinations (homozygous and heterozygous genotypes) as high-activity alleles and genotypes, respectively [26]. Du et al. [23] also grouped these alleles into a single allele group for analysis against the low-activity allele 3R. Although 5R is rare, it is controversial regarding its transcriptional efficiency (see the Introduction), so subdivisions, such as these might contribute to a sample reduction for each analysis, influencing the frequency estimates [23,24,25,26,40]. The 3R/3R and 3R* genotypes frequency obtained by Du et al. [23] in a Canadian MDD patient sample (Hamilton Depression Rating Scale, HAM-D: males, 22.9 ± 3.5; females, 23.6 ± 3.2) was 37.2%, close to the 40% reported by Schulze et al. [40] for recurrent depression (DSM-IV) in a German population.

Most control groups had the 3R/3R and 3R* genotype frequencies higher than those of the MDD groups (Table 1 and Table 2), a fact that may represent an intriguing finding as the 3R allele presence may represent a protective factor for MDD. This protection might be due to the lower MAOA transcriptional efficiency [17,41] and enzymatic activity [18] that the 3R allele confers, contributing to the maintenance of central serotonin levels [33].

The 4R allele is highly active [18], which may help reduce serotonin levels in the central nervous system, possibly modulating the risk of depression [27]. Lung et al. [27] explored the *MAOA uVNTR* variant association with MDD and suicide attempts. The authors found that the 4R allele frequency was significantly higher in men with the disorder compared to male community controls (X^2^ = 4.182, *p* = 0.041), irrespective of if they had attempted suicide, and that 4R* males had a 1.586 (Odds Ratio Confidence Interval, OR CI: 1.019–2.467) times higher risk of developing depression than 3R* males. Although their logistic regression analysis did not confirm this association, the 3R allele was more present in male community controls (60.98%, Table 1 and Table 2). The 4R allele also appeared to indirectly affect suicide attempts associated with depressive symptoms (structural equation modeling, SEM ~ MAOA-depression: b = −0.12, *p* = 0.031; depression-suicide: b = 0.32, *p* < 0.001), conferring a vulnerability to suicide in MDD men [27]. Similarly, Sanabrais-Jiménez et al. [36] investigated *SLC6A4 (5HTTLPR/rs25531), DRD2 (rs6275), COMT (rs4680)*, and *MAOA uVNTR* polymorphisms’ influences on suicide attempts in Mexican adolescents diagnosed with MDD (DSM-IV major depression). However, unlike Lung et al. [27], the *MAOA uVNTR* variant did not correlate with suicide attempts in these patients (females—genotype: X^2^ = 0.86, *p* = 0.64; allele: X^2^ = 0, *p* = 1 | males—X^2^ = 0.008, *p* = 0.92)—results compatible with those found by Hung et al. [42] in his metanalysis.

Du et al. [23] investigated *MAOA* (*EcoRV* and *uVNTR*) polymorphisms’ connection with MDD and depressive symptoms and found that *MAOA uVNTR* did not associate with MDD, although it trended towards significance in male patients (*p* = 0.055). Du et al. [23] did find a highly significant linkage disequilibrium (a nonrandom association of alleles of different loci) between *MAOA*’s *uVNTR* and *EcoRV* variants (D’_controls_ = −0.79; *p* < 0.0001; D’_patients_ = 0.84; *p* < 0.0001). They also found that the *EcoRV* allele with the EcoRV site present and *MAOA uVNTR*’s 3R allele significantly correlated with depression in males better than any of the polymorphic alleles alone (*p* = 0.008; OR = 2.5, OR CI = 1.3–4.8) and with higher insomnia scores compared to other haplotype carriers (HAM-D clusters compared by unpaired Student *t*-test, t = 2.7, *p* = 0.008), even after correction for multiple testing (*p* = 0.048 in both cases) [23]. Huang et al. [30] also found a strong linkage disequilibrium between these *MAOA* variants but found no significant differences in the haplotype frequencies in total MDD patients (DSM-IV major depression; HAM-D ≥ 18) or MDD clinical subgroups (18 ≤ HAM-D ≤ 24: moderate MDD; HAM-D > 24: severe MDD; MDD with and without a first-degree family member with a history of MDD or bipolar disorder) versus controls (*p* > 0.05, data not shown).

Rivera et al. [26] found an association between high-activity alleles (3.5R; 4R; and the supposedly 5R; see the Introduction) and MDD in female Spanish patients (*p* = 0.048), similarly to Huang et al. [30]; who found a weak association between severe MDD group and its genotypic frequencies among female Taiwanese participants (*p* = 0.041). However, like with Lung et al. [27]; the association was not maintained after multiple logistic regression analyses; suggesting that the *MAOA uVNTR* variant may not play a central role in the risk of developing MDD [30]. Yu et al. [33] also assessed the association between the *MAOA uVNTR* polymorphism and MDD (DSM-IV major depression; HAM-D-1967 ≥ 18) and found that the 4R allele was more common in MDD women (X^2^ = 6.93; df = 1; *p* = 0.009) and MDD men (X^2^ = 6.27; df = 1; *p* = 0.015) compared to the control group (mostly composed of medical staff) [33]. These findings suggest a non-gender-specific effect on MDD given the increased 4R allele presence in the MDD group (X^2^ = 12.48; df = 1; *p* < 0.001), even considering X-inactivation at the MAOA locus (X^2^ = 10.7,2; df = 1; *p* = 0.001) [33].

Like with the Brazilian population; there is a lack of studies analyzing the *MAOA uVNTR* variant effect on patients diagnosed with MDD. Other studies trying to relate this variant to other monoamine-metabolism-affected neurological/psychopathological disorders: such as aggression and antisocial behavior; mood disorders; schizophrenia; autism spectrum disorders; substance use disorders; and even Alzheimer’s disease; were found in the literature [43,44,45]. For instance: the high-activity alleles’ presence correlated with clinical improvement of opposing symptoms in male children and adolescents with Attention Deficit Hyperactivity Disorder using methylphenidate (*p* < 0.001) [46]; the occurrence of bilateral tonic-clonic seizures in patients with epilepsy (*p* = 0.032) [47]; and increased consumption of lipid-dense foods in preschool children (*p* = 0.009) [48]. At the same time: the low-activity allele correlated with an early-age onset of alcohol dependence (*p* = 0.01) and a more considerable amount of antisocial personality symptoms after age 15 (*p* = 0.02) [49].

Such associations and those with MDD seem to indicate this polymorphism’s influence on monoaminergic metabolic pathways. The WHO estimates a 5.8% prevalence of depressive disorders in Brazil [1], a country composed of a population with a heterogeneous genetic makeup—Native Americans, Europeans, and Africans [43]. This heterogeneity highlights the need for further studies evaluating the behavior of the *MAOA uVNTR* variant’s genotypic distribution and allelic frequencies in multiracial populations, together with more single ethnicity populations studies, to fill the knowledge gaps and build better risk prediction models.

### 4.2. MAOA uVNTR Genetic Variant and Cortical Thickness

Neuroimaging studies are essential when analyzing mood disorders as they help identify neuroanatomical alterations in brain regions brought on by the disorder, its treatment, and other factors [4,34]. Suh et al.’s [4] systematic review and meta-analysis analyzed neurobiological differences between MDD (medicated and medication-naïve) patients (*n* = 1073) and healthy controls (*n* = 936). They found that MDD patients have a significant cortical thinning in the bilateral medial orbitofrontal cortex (OFC), left pars opercularis, and left calcarine fissure/lingual gyrus, as well as a significant thickening in the left supramarginal gyrus compared to healthy controls. However, it is still unclear what factors may contribute to these alterations in MDD [4,34].

The OFC seems to regulate mood, reward-guided behavior, and impulse control; thus, damage to this structure might create deficits in an individual’s emotional and social regulation and mood lability [34]. With the majority of imaging genetics studies conducted on *MAOA uVNTR* polymorphism reporting its genotype relevance to OFC structure in healthy controls, Won et al. [34] compared this variant effect on OFC thickness of Korean female MDD patients (medication-naïve; HAM-D: 20.96 ± 5.09) and healthy controls (HAM-D: 2.41 ± 2.20). Similar to Suh et al. [4]’s results, Korean MDD patients presented cortical thinning in the bilateral medial (left—F(1,71) = 8.117, *p* = 0.006; right—F(1,71) = 21.795, *p* < 0.001) and in the right lateral (F(1,71) = 9.932, *p* = 0.002) OFC compared to healthy controls, regardless of their *MAOA uVNTR* genotype (*MAOA uVNTR* high-activity allele carriers, 3R/4R and 4R/4R, or *MAOA uVNTR* low-activity allele carriers, 2R/3R and 3R/3R).

Although these results suggest that OFC structural alterations are associated more with MDD pathophysiology than *MAOA uVNTR* polymorphism, this lack of correlation might be due to the samples’ composition, as *MAOA uVNTR* genotype-dependent OFC structural changes have usually been observed in males but not females [43,50,51,52,53,54].

### 4.3. MAOA uVNTR Genetic Variant and Pharmacotherapy

Drugs with a mechanism of action that increases neurotransmitter levels in the synaptic cleft by inhibiting specific transporters have an antidepressant effect, such as monoamine oxidase inhibitors (MAOI) and other drugs [23,45,55,56]. In their MDD patients study, Yu et al. [33] observed that female (3R/3R genotype) patients responded better to 4-week fluoxetine (an antidepressant belonging to the Selective Serotonin Reuptake Inhibitors, SSRI, class) treatment than those carrying the 4R allele (3R/3R: 46.9 ± 19.0%, 3R/4R: 37.3 ± 19.8%, 4R/4R: 34.1 ± 19.4%, respectively, *p* = 0.024) according to the change in the Hamilton Depression Rating Scale (HAM-D). Contrarily, the difference in male (3R* genotype) patients’ responses was nonsignificant (3R*: 39.1 ± 21.0%, 4R*: 31.0 ± 23.6%, respectively, *p* = 0.081), possibly indicating a biological-sex-specific effect. Although a multiple linear regression analysis initially confirmed these findings (*p* = 0.010; r^2^ = 0.049), the association was not maintained after correction for multiple testing stratified by biological sex [33]. Differently, Yoshida et al. [55] could not establish an association between plasma drug levels and *MAOA uVNTR* polymorphism’s genotypes after a 6-week Fluvoxamine (SSRI) treatment in a Japanese population with MDD (*p* = 0.92).

Tzeng et al. [56], in turn, investigated the association between *MAOA uVNTR* polymorphism’s genotypes and their response to the Mirtazapine, an atypical antidepressant belonging to the tetracyclic piperazine-azepine class, in a Taiwanese sample and found an association between medication use and treatment variations according to the genotype. In their study, 3R allele carriers had a better therapeutic response and higher remission rates after a 7-week Mirtazapine treatment. Noticeably, 3R allele carriers also had a lower dropout rate; however, the sample size was small (*n* = 58), which might generate a bias [56].

### 4.4. Quality Assessment and Limitations of Selected Articles

Genomic-wide association studies investigate the association of genotypic variations and characteristics expressed by individuals in different populations [57], including susceptibility/resistance to certain diseases and disorders. These studies have great applicability; among them is the implementation of these possible connections to construct analytical designs/models to predict the genetic risk of several diseases [58]. Studies to better understand genetic and environmental factors are increasingly important in scientific areas, specifically health. In this sense, the Genetic Risk Prediction Studies (GRIPS) guideline ensures a higher methodological quality of a study, enabling the minimization of biases that can compromise the interpretation of results [28,29].

We employed the GRIPS guideline, composed of 25 items, to assess the quality of the articles selected to integrate this systematic review (Appendix A). Criteria chosen to analyze the quality of methods section were ten items (two sub-items), results: seven items, and discussion: three items, totaling 20 items (and two sub-items). We considered the study quality adequate if it had 75% (15) of the items, and all articles included met this criterion. We considered the study quality adequate if it had at least 75% (15) of the items, and all articles included in this review met this criterion. All articles [23,26,27,30,33,34,36] described the participants’ eligibility criteria and sources and methods of selecting participants. They also made a generalized discussion and, when pertinent, revealed the importance of their study. Nevertheless, for instance, 71.4% [23,27,33,34,36] examined their limitations, one (14,3%) did not report their population’s demographic and clinical characteristics [30], and none specified how they handled missing data in the analysis.

High-quality studies must be prepared transparently to facilitate the interpretation of the results’ validity and assess their relevance. Observations in different populations are essential to generalize the research and help minimize the existing heterogeneity factor [59]. In this way, these studies allow the information to be extrapolated to different contexts and help construct an integrative data analysis model for eventual practical application [28].

## 5. Conclusions

Although some evidence suggests that the *MAOA uVNTR* variant’s alleles are related to MDD manifestation, there is still no concrete accord on this association. In general, the *MAOA uVNTR* variant’s 3R and 4R alleles had the highest frequency in the populations studied, whereas the 2R, 3.5R, 5R, and 6R alleles, in most studies, were considered rare or not found at all. Interestingly, the low-activity allele 3R presented a higher frequency in most control groups, and the reverse was true for the 4R allele in the studies when not grouped with other alleles. Nevertheless, the number of retrieved studies that met our inclusion criteria was small, representing a limiting factor in this systematic review. The different allele groupings used in some of the articles’ analyses also increased the difficulty in interpretation as some analyzed 3R/3R and 3R* genotype and allelic frequencies against 4R/4R and 4R*, while others grouped low-activity genotypes and alleles (3R) against high-activity ones (3.5R and 4R, that sometimes included 5R). Another limitation was the few represented populations.

Depression is a heterogeneous and multifactorial disease with both genetic and environmental factors influencing its development. Sex-specific effects and possible interactions with other polymorphisms and genes (e.g., linkage disequilibrium), as well as epigenetic and environmental factors, must be taken into account to understand this polymorphism influence on MDD better, thus contributing to the elucidation of the mechanisms related to this disorder—further studies on a larger scale and in other populations will help fill these gaps.

## Figures and Tables

**Figure 1 cells-11-03267-f001:**
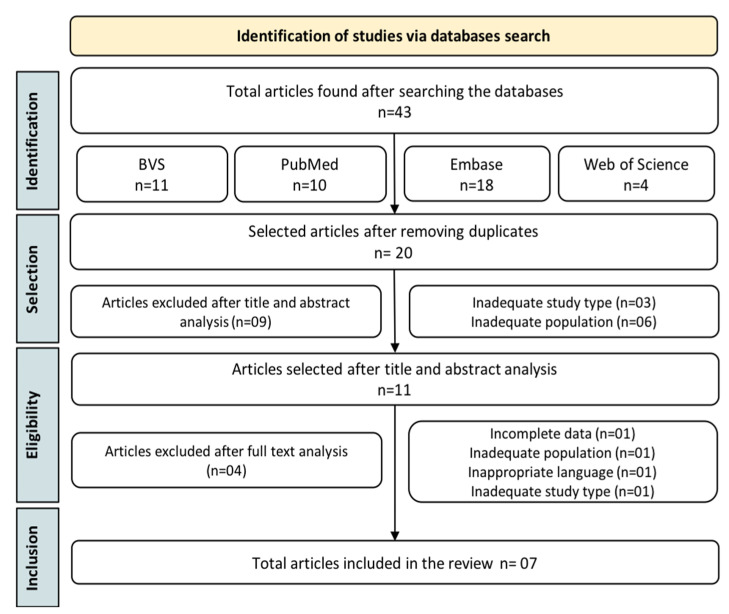
Flowchart outlining the steps adopted in the articles’ selection.

**Table 1 cells-11-03267-t001:** The comparison of different populational studies that investigated the *MAOA uVNTR* polymorphism associated with Major Depression Disorder (MDD).

Author	Year	Title	Country	Objective	Sample	Genetic Variant	Genotype Frequency (3R/3R and 3R*)	Laboratorial Test	Results	*p*-Value ^a^ (3R/3R and 3R*)
Huang et al. [30]	2009	Association of monoamine oxidase A (MAOA) polymorphisms and clinical subgroups of major depressive disorders in the Han Chinese population	Taiwan	Investigate whether the *MAOA* gene’s *uVNTR* and *EcoRV* variants are associated with MDD or other MDD clinical subgroups in a Han Chinese Taiwanese population.	MDD = 277 Control = 308	*MAOA EcoRv* ^@^ *(rs1137070)* *MAOA uVNTR*	MDD: F = 40.6% (*n* = 69); M (3R*) = 57.9% (*n* = 62); Total: 47.29% (*n* = 131). Control: F = 32.4% (*n* = 36); M (3R*) = 62.9% (*n* = 124); Total: 51.95% (*n* = 160).	PCR per Zhu et al. [31]’s and PCR-RFLP per Breakfield et al. [32]‘s protocols	Among females, the MAOA uVNTR genotypic frequencies showed a weak association between severe MDD and control (*p* = 0.041), but not among males (*p* > 0.1). However, after multiple logistic regression analyses, the association was not maintained.	F (genotype) = 0.158 M (allele) = 0.393
Du et al. [23]	2004	MAO-A gene polymorphisms are associated with major depression and sleep disturbance in males	Canada	Investigate whether the *MAOA* gene’s *uVNTR* and *EcoRV* variants are associated with MDD, specific depressive symptoms, or both.	MDD = 191 Control = 233	*MAOA EcoRv*@ *(rs1137070)* *MAOA uVNTR*	MDD: F = 35% (*n* = 79); M (3R*) = 43.6 (*n* = 34); Total: 37.17% (*n* = 113). Control: F = 31.4% (*n* = 81); M (3R*) = 29.8% (*n* = 31); Total: 30.94% (*n* = 112).	PCR per Deckert et al. [21]’s protocol	The alleles’ distribution showed no statistical difference between patients and the control group, but in male patients, it tended toward significance (*p* = 0.055), suggesting that there may be a biological-sex-base relationship between the polymorphism and MDD.	F (allele) = 0.4 M (allele) = 0.055
Rivera et al. [26]	2009	High-Activity Variants of the uMAOA Polymorphism Increase the Risk for Depression in a Large Primary Care Sample	Spain	Clarify the *MAOA uVNTR* polymorphism association with depression, in a large and well-characterized representative population, in primary care.	MDD = 243 Control = 980	*MAOA uVNTR*	MDD: F = 11% (*n* = 23); M = 26% (*n* = 9); Total: 13.17%(*n* = 32). Control: F = 17% (*n* = 113); M = 34.5% (*n* = 107); Total: 22.45% (*n* = 220).	PCR	*MAOA uVNTR*’s high-activity (3.5R, 4R, 5R, and their combinations) variants in females correlated with MDD (*p* = 0.048) but not in males (*p* = 0.229).	F = 0.024 M = 0.229 Total (M + F) = 0.0014
Yu et al. [33]	2005	Association Study of a Monoamine Oxidase A Gene Promoter Polymorphism with Major Depressive Disorder and Antidepressant Response	China	Study the *MAOA uVNTR* polymorphism’s association in a Chinese population and its influence on MDD patients’ antidepressant therapeutic response (fluoxetine treatment for four weeks observation).	MDD = 230 Control = 217	*MAOA uVNTR*	MDD: F = 25.6% (*n* = 34); M (3R*) = 48.4% (*n* = 46); Total: 35.09% (*n* = 80). Control: F = 44.5% (*n* = 49); M (3R*) = 66.0% (*n* = 68); Total: 54.93% (*n* = 117).	PCR	Female 3R/3R genotype carriers responded better to 4-week fluoxetine treatment than those with 4R alleles (*p* = 0.024). Furthermore, 4R allele carriers are more common in the MDD group, regardless of biological sex.	F (genotype) = 0.008 M (allele) = 0.015
Lung et al. [27]	2011	Association of the MAOA promoter uVNTR polymorphism with suicide attempts in patients with major depressive disorder	Taiwan	Investigate the *MAOA uVNTR* variant’s role in MDD, suicide attempts, or both.	MDD = 379 Control = 420	*MAOA uVNTR*	MDD^∞^: F = 36.08% (*n* = 105); M (3R*) = 48.96% (*n* = 94); Total: 41.20%(*n* = 199). Control^∞^: F = 35.87% (*n* = 99); M (3R*) = 59.63% (*n* = 130); Total: 46.36%(*n* = 229)	PCR	The high activity 4R allele seems related to depression, as it was more common among male participants with MDD than those without (*p* = 0.041); it also seems to affect suicide attempts associated with depressive symptoms in males indirectly.	F (allele) = 0.639 M (allele) = 0.041
Won et al. [34]	2016	Regional cortical thinning of the orbitofrontal cortex in medication-naïve female patients with major depressive disorder is not associated with MAOA-uVNTR polymorphism	Korea	Investigate the difference in orbitofrontal cortex thickness between medication-naïve female MDD patients and healthy controls and the *MAOA uVNTR* genotype’s influence on orbitofrontal cortex thickness in depression.	MDD = 31 Control = 43	*MAOA uVNTR*	MDD: F = 41.93% (*n* = 13). Control: F = 39.53% (*n* = 17).	Genotyping per Manor et al. [35]’s protocol	MDD patients presented thinning in bilateral medial (*p* < 0.01) and in the right lateral (*p* = 0.002) orbitofrontal cortex compared to healthy controls, regardless of their *MAOA uVNTR* genotype.	F (allele) = 0.542
Sanabrais-Jiménez et al. [36]	2021	Association study of Catechol-O-Methyltransferase (COMT) rs4680 Val158Met gene polymorphism and suicide attempt in Mexican adolescents with major depressive disorder	Mexico	Analyze the association between *SLC6A4, DRD2, COMT,* and *MAOA* genes and suicide attempts in Mexican adolescent MDD patients.	MDD = 197	*MAOA uVNTR*	MDD^∞^: F = 18.5% (*n* = 23); M (3R*) = 41.2% (*n* = 28); Total: 26.56% (*n* = 51).	Genotyping per Camarena et al. [37]’s protocol	The *MAOA uVNTR* variant did not correlate with suicide attempts in MDD patients (*p* > 0.05).	F (genotype) = 0.64 M (allele) = 0.92

Note: 3R*—male hemizygotes for the 3R allele; F—females and M—males; MDD^∞^: value calculated by uniting different MDD subgroups (e.g., with/without suicide attempt) into one; Control^∞^: value calculated by uniting control subgroups (e.g., with/without suicide attempt) into one. ^a^ Statistical test: Chi-square (X^2^) test. ^@^*MAOA EcoRV* (rs1137070; 1460C > T) genetic variant affects its enzyme activity and its alleles are differentiated by the absence (−; C) or presence (+; T) of the EcoRV restriction length polymorphism site on exon 14 (position 1460).

**Table 2 cells-11-03267-t002:** Comparison between *MAOA uVNTR* variant’s 3R/3R (female homozygote) and 3R* (male hemizygote) genotype frequencies^@^ in patients with Major Depression Disorder (MDD) and healthy controls in the studies reviewed per continent.

Continents	Works	MDD Group			Control Group		
		Female 3R/3R % (n)	Male 3R* % (n)	Total 3R/3R + 3R* % (n)	Female 3R/3R %(n)	Male 3R* % (n)	Total 3R/3R + 3R* % (n)
Asian	Yu et al. [33] 2005 (China)	25.6 (34)	48.4 (46)	35.1 (80)	44.5 (49)	66 (68)	54.93 (117)
	Huang et al. [30] 2009 (Taiwan)	40.6 (69)	57.9 (62)	47.29 (131)	32.4 (36)	62.9 (124)	51.95 (160)
	Lung et al. [27] 2011 (Taiwan)	36.08 (105)	48.96 (94)	41.2 (199)	35.87 (99)	59.63 (130)	46.36 (229)
	Won et al. [34] 2016 (Korea)	41.93 (13)	-	41.93 (13)	39.53 (17	-	39,53 (17)
American	Du et al. [23] 2004 (Canada)	35 (79)	43.6 (34)	37.17 (113)	31.4 (81)	29.8 (3)	30.94 (112)
	Sanabrais-Jiménez et al. [36] 2021 (Mexico)	18.5 (23)	41.2 (28)	26.56 (51)	-	-	-
European	Rivera et al. [26] 2009 (Spain)	11 (23)	26 (9)	13.17 (32)	17 (113)	34.5 (34.5)	22.45 (220)

* Male hemizygotes for the 3R allele. ^@^ Individual and combined 3R/3R and 3R* frequencies were calculated from all frequencies of all genotypes for each data group [(genotype of interest) ÷ (all genotypes) × 100].

## Data Availability

No new data were created in this study, and the data presented in this review are available in its tables (Table 1, Appendix A) and the reviewed original articles.

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
