# Peer review of "MAOA uVNTR Genetic Variant and Major Depressive Disorder: A Systematic Review"

_cells, 2022, doi:10.3390/cells11203267_

Round 1

Reviewer 1 Report

The authors have conducted a systematic review including studies investigating the role of the MAOA VNTR polymorphism in major depressive disorder. The review has been conducted in accordance with PRISMA guidelines. The rationale of the review is well explained and the topic is indeed relevant.

The authors do not report a PROSPERO number but they state it is pending due to COVID-19 reviews prioritization.  In any case, the search strategy, the inclusion criteria as well as the evauation process of retrieved articles are well explained. I only have minor comments.

- In Table 1, it was not clear to me what the sentences "Genotyping per Manor et al. 2002" and "Genotyping per Camarena et al. 2012" in the laboratorial test column, for the last two articles mean. Also, these articles (Manor et al., 2002 and Camarena et al., 2012) do not seem to be present in the reference list (I'm not sure whether they should).

- Page numbers should be rechecked as many have wrong numbers

- Figure 2 is numbered as Figure 1. Also, this figure doesn't seem very useful to me. 

- Figure 3 example legend should be removed. Same for formatting guidelines for the references. 

- Figure 3 is a bit difficult to read. Maybe the A and B panels could be split in different figures. 

Author Response

You can find the reply attached below. Thank you. 

Reviewer 2 Report

In the manuscript "MAOA uVNTR genetic variant and Major Depressive Disorder: A Systematic Review" by Castro et al, the authors present a review of the current literature of genetic association studies investigating this polymorphism in the context of MDD.

While the manuscript could be of possible interest for the field, there are several concerns and points that should be addressed by the authors:

1. Manuscript needs help with English language, some sentences / phrases are difficult to understand and may be misinterpreted.

2. The authors should avoid making statements that could indicate a potential causality / causal relationship between the polymorphism and MDD or other outcomes as such conclusions cannot be made from the type of research (genetic association studies) included in this review.

3. When explaining the current evidence on the MAOA uVNTR polymorphism the authors should provide more detailed and specific information including references how “low-activity” / “high activity” etc. was determined in previous studies including the models / populations studied (cells, animals, human participants). Currently this part is quite vague and confusing for readers particularly for those not familiar with this topic.

4. In their hypotheses the authors include that carrying the 4R allele may increase susceptibility for MDD, however, they then state that the systematic review focuses on 3R?

5. The MAOA gene is located on the X chromosome – the authors do not provide specific information on this in the introduction and they do not provide information if there have been different findings in individuals with male / female sex related to this.

6. The authors should include information if a previous review on this topic has been conducted / published and if so, when this was done.

7. As the authors refer to Prospero registration, information should be provided when this was done, if pending information on when study design was submitted / finalized.

8. The authors introduce “3R*” in line 100 but do not explain what this means. The authors introduce “EcoRv” for the first time in table one but do not explain this term / acronym.

9. The authors used the search term "MAOA uVNTR" could it be possible that the polymorphism was not always reported as " uVNTR" in previous, particularly in earlier publications?

10. Very important scientific aspects of research studies / genetic studies were not extracted from the studies / are not reported in the manuscript:

- Why did the authors not extract and include data on ethnicity and/or race? - The authors have not extracted / included information on how MDD was assessed / diagnosed or how healthy controls were assessed to identify them as “healthy”

- The authors did also not include information on what statistical tests / approaches were used in the studies.

- Also, it is very important for genetic association studies if results were controlled for multiple testing – this was not extracted or reported in the manuscript.

11. The authors indicate that "interventional" studies were included although they state that the review focuses on association studies of the MAOA polymorphism and susceptibility for MDD - thus, the rationale for including interventional studies is not clear. The authors report various results in Table 1 including if polymorphism was associated with response to treatment or suicide attempts which is not within the objective stated by the authors in the introduction of the manuscript. Overall, rationale and objective of the study are not clearly described or do not match with the studies included / results presented. 

12. In the methods and results section the authors refer to “eligibility criteria” or “previously established inclusion and exclusion criteria”, however these are not clearly described and/or presented in the manuscript.

13. In addition, the actual search terms should be included / presented including all details (this could be done in supplementary information).

14. In the PRISMA diagram the authors describe that one paper was excluded due to “inappropriate language” or “inadequate population” or “inadequate study type”, the authors may consider changing this to excluded due to “study type”, “study population”, “language of article” etc.

15. There are two Figure 1.

16. The second Figure 1 is unnecessary – this information can be described in one short sentence.

17. Table 1: “MDD∞: value calculated with MDD groups united into one; Control∞: value calculated with control groups united into one” – it is unclear what the authors are referring to here.

18. Figure 3 is confusing / difficult to understand – are there missing bars?

19. The authors miss to provide a structured critical synthesis and summary in the current version of the manuscript, which is the specific aspect and reason of a systematic review, particularly with respect to what appears to be the main objective of this work, i.e. association of  MAOA uVNTR with MDD. (This is to some extent included in the discussion section but is difficult to follow)

20. The authors refer to a risk of bias assessment in the methods section but do not present this in the results section. This has been included in the supplementary information, however, the authors only applied part of the tool for RoB assessment but rationale for doing this is not provided. 

21. The discussion section is lengthy, particularly when considering the content and overall limited findings / information and length of the results.

22. Supplementary table 1 – it is unnecessary to list duplicate articles.

23. Supplementary table 1: See comment 14. above regarding “reason for exclusion”.

Author Response

(The authors gave the same response as above.)
